# Comparing the Differences in Adverse Events among Chimeric Antigen Receptor T-Cell Therapies: A Real-World Pharmacovigilance Study

**DOI:** 10.3390/ph17081025

**Published:** 2024-08-05

**Authors:** Zihan Guo, Yunlan Ding, Mengmeng Wang, Qing Zhai, Jiyong Liu, Qiong Du

**Affiliations:** 1Department of Pharmacy, Fudan University Shanghai Cancer Center, Shanghai 200032, China; gzhmay@126.com (Z.G.); 18851100565@163.com (Y.D.); cassie0510@163.com (M.W.); zhaiqing63@126.com (Q.Z.); 2Department of Oncology, Shanghai Medical College, Fudan University, Shanghai 200032, China

**Keywords:** CAR T-cell therapy, adverse event, FAERS, real world, pharmacovigilance

## Abstract

In this study, we compared the similarities and differences in adverse events (AEs) among CAR T-cell products through signal mining via the FDA Adverse Event Reporting System (FAERS) and identified unknown AEs to provide a reference for safe clinical medication. Data from the FAERS database spanning from the fourth quarter of 2017 to the first quarter of 2024 were extracted. Signals were identified using the reporting odds ratio (ROR) method and the Medicines and Healthcare Products Regulatory Agency (MHRA) method. A total of 11,386 AE reports related to six CAR T-cell products were selected. The top three categories of AEs reported were nervous system disorders, immune system disorders, and general disorders and administration site conditions. However, there were variations in the AE spectra among the different CAR T-cell products. The BCMA-targeting drugs idecabtagene vicleucel (Ide-cel) and ciltacabtagene autoleucel (Cilta-cel) were found to be associated with parkinsonism, which were not observed in CD19-targeting drugs. Tisagenlecleucel (Tisa-cel) and axicabtagene ciloleucel (Axi-cel) exhibited cerebrovascular accident-related AEs, graft versus host disease, and abnormal coagulation indices. Cilta-cel was associated with cerebral hemorrhage, intracranial hemorrhage, cranial nerve disorder, and facial nerve disorder. Cardiopulmonary toxicity, including hypoxia, tachypnoea, cardiorenal syndrome, and hypotension, exhibited strong signal intensities and considerable overlap with CRS. The number of positive signals for cardiopulmonary toxicity associated with drugs targeting CD-19 is greater. Clinicians should assess patients prior to medication and closely monitor their vital signs, mental status, and laboratory parameters during treatment.

## 1. Introduction

Immunotherapy has become a research hotspot in the field of hematological malignancies in recent years. With advancements in genetic engineering and molecular biology and a deeper understanding of tumor pathogenesis, chimeric antigen receptor T-cell (CAR T-cell) therapy has seen widespread clinical application. Extensive clinical research has demonstrated that CAR T-cell therapies are highly effective in treating relapsed/refractory (RR) B-cell hematological malignancies [1,2,3]. Specifically, the complete response of patients with acute lymphoblastic leukemia (ALL) ranges from 70% to 90% [4,5,6,7,8,9,10,11,12,13], that of patients with lymphoma ranges from 50% to 70% [14,15,16,17], and that of patients with multiple myeloma (MM) ranges from 50% to 98% [18,19,20,21,22,23,24,25]. In 2017, two CD19-targeted CAR T-cell products, tisagenlecleucel (Tisa-cel) and axicabtagene ciloleucel (Axi-cel), were approved by the U.S. Food and Drug Administration (FDA) for the treatment of relapsed/refractory ALL and lymphoma [26,27]. Subsequently, four additional CAR T-cell products were approved by the FDA: lisocabtagene maraleucel (Liso-cel) and brexucabtagene autoleucel (Brexu-cel), which target CD19, and idecabtagene vicleucel (Ide-cel) and ciltacabtagene autoleucel (Cilta-cel), which target B-cell maturation antigen (BCMA) [28,29,30]. With the increasing clinical application of CAR T-cell therapy, treatment-related side effects have garnered significant attention. In addition to cytokine release syndrome (CRS) and immune effector cell-associated neurotoxicity syndrome (ICANS), a variety of other AEs, such as hematological disorders, infections, and cardiovascular diseases, have been observed in clinical trials. In some instances, these AEs can be fatal [17,31,32,33], posing considerable challenges to routine clinical diagnosis and treatment. At present, the FDA has approved six CAR T-cell products, four of which target CD19 and two of which target BCMA, but no relevant studies have analyzed and compared the adverse reactions of CAR T-cell products with different targets.

Isolated clinical trials and their systematic reviews present the highest quality of evidence and are the basis for guidelines issued by healthcare organizations. However, the evaluation of entire profiles of rare AEs derived from clinical trials is difficult owing to their stringent diagnostic standards and selection criteria, relatively small sample sizes, and limited follow-up time. The FAERS database, one of the largest pharmacovigilance databases, with a large quantity of reported AEs and patient information, could provide data to verify and supplement the findings of clinical trials. In this study, we compared the similarities and differences in AEs among six CAR T-cell products through signal mining via the FAERS database and identified unknown AEs to provide a valuable reference for the selection and long-term management of CAR T-cell therapy.

## 2. Results

### 2.1. Characteristics of CAR T-Cell Reports in the FAERS Database

A total of 11,616,357 reports were retrieved in the FAERS database from the inception up to the first quarter (Q1) of 2024, and 11,386 reports were associated with FDA-approved CAR T-cell products after deduplication (2689 involving Tisa-cel, 5602 involving Axi-cel, 1010 involving Brexu-cel, 348 involving Liso-cel, 654 involving Ide-cel, and 1083 involving Cilta-cel) (Figure 1). The specific demographic and clinical details are provided in Table 1. Among these reports, 29.9% were female and 48.7% were male. They were mainly aged 18 to 64 years (33.4%). From the perspective of reporting sources, health professionals reported the most cases (80.8%). Most AE cases were reported in North America, accounting for 63.4% of all cases. This was followed by Europe (20.3%) and Asia (4.3%). Furthermore, a significant proportion of patients (92.1%) experienced serious outcomes, including hospitalization, death, and life threat. The major ADRs overlapped with CRS.

### 2.2. CAR T-Cell Therapy-Associated Adverse Events

The AE signals were classified according to the system organ class (SOC), and the proportion of AE-positive signals related to CAR T-cell therapy is shown in Figure 2. Overall, nervous system disorders, immune system disorders, and general disorders and administration site conditions are the top three categories of AEs reported for CAR T-cell therapy. Notably, Tisa-cel has a greater incidence of AEs related to various laboratory tests, Ide-cel has a greater incidence of immune system-related AEs, and Brexu-cel has a greater incidence of nervous system-related AEs.

#### 2.2.1. Common Adverse Events in Key Organs

The results of the detection of common AE signals associated with CAR T-cell therapy in key organs are shown in Table 2, and differences in the AE profiles were found among the CAR T-cell products. The details are as follows:

Nervous system disorders: ICANS and neurotoxicity are the most common AEs, showing strong signal strength across all drugs. Axi-cel exhibited the strongest signal for ICANS (ROR = 1198.80), while Liso-cel showed the strongest signal for neurotoxicity (ROR = 236.75). Tisa-cel and Axi-cel also exhibited cerebrovascular accident-related AEs. Cilta-cel was associated with cerebral hemorrhage (ROR = 3.81) and intracranial hemorrhage (ROR = 18.47) and exhibited strong signals in cranial nerve disorder (ROR = 991.24), facial nerve disorders (ROR = 727.22), Bell’s palsy (ROR = 253.07), facial paralysis (ROR = 95.69), and Guillain–Barre syndrome (ROR = 19.92). The BCMA-targeting drugs Ide-cel (ROR = 13.00) and Cilta-cel (ROR = 168.90) were found to be associated with parkinsonism, which were not observed in CD19-targeting drugs. There were more AEs in the nervous system, and only AEs with an ROR greater than or equal to 5 are listed in the table.

Immune system disorders: CRS was the most common AE, followed by hypogammaglobulinemia and hemophagocytic lymphohistiocytosis (HLH). Axi-cel had the strongest signal for CRS (ROR = 676.42), Tisa-cel had the strongest signal for hypogammaglobulinemia (ROR = 151.64), and Cilta-cel had the strongest signal for HLH (ROR = 75.22). Additionally, Tisa-cel and Axi-cel were found to cause graft versus host disease. Axi-cel, Ide-cel, and Cilta-cel were also associated with immunodeficiency.

Infections and infestations: Bacterial infection was the most common type of infection, with Cilta-cel showing relatively strong signals of bacterial (ROR = 7.16), fungal (ROR = 8.17), and viral (ROR = 2.63) infections.

Investigations: Decreased white blood cell count and decreased platelet count were the most common AEs, with Tisa-cel showing relatively strong signals. Tisa-cel, Axi-cel, Brexu-cel, Liso-cel and Cilta-cel all exhibited strong signals for increased serum ferritin. Additionally, Tisa-cel and Axi-cel also observed abnormal coagulation indices.

#### 2.2.2. Adverse Events of Special Interest

Respiratory, thoracic, and mediastinal disorders: Hypoxia, respiratory failure, tachypnea, and pleural effusion were the most common AEs, and strong signals were observed in several CAR T-cell products. Tisa-cel, Axi-cel, and Brexu-cel also exhibited atelectasis, which was not mentioned on the drug labels. Additionally, Tisa-cel was associated with respiratory distress (ROR = 4.08), pulmonary hemorrhage (ROR = 5.29), organizing pneumonia (ROR = 5.00), lung consolidation (ROR = 5.99), and pharyngeal hemorrhage (ROR = 10.64). Axi-cel was associated with pulmonary hemorrhage (ROR = 3.87), lung consolidation (ROR = 4.07), lung infiltration (ROR = 3.56), and pulmonary alveolar hemorrhage (ROR = 2.98). Ide-cel was linked to laryngeal edema (ROR = 30.63). Cilta-cel also detected respiratory distress (ROR = 6.14). Although these AEs are uncommon, they still necessitate clinical attention (Table 3).

Cardiac disorders: Tachycardia and arrhythmia were the most common AEs. Additionally, Tisa-cel (ROR = 44.41) and Axi-cel (ROR = 150.61) were associated with cardiorenal syndrome. Tisa-cel was also linked to pericardial effusion (ROR = 2.41), left ventricular dysfunction (ROR = 5.27), mitral valve disease (ROR = 3.78), and aortic valve incompetence (ROR = 4.71). Axi-cel also showed signals of cardiomyopathy (ROR = 4.43) and cardiopulmonary failure (ROR = 4.41) (Table 3).

Vascular disorders: Hypotension was the most common AE, showing strong signals across several drugs. Shock was also detected for Tisa-cel (ROR = 3.20), Axi-cel (ROR = 2.87), Brexu-cel (ROR = 4.31), and Ide-cel (ROR = 4.41), which was not mentioned on the drug labels. Additionally, Tisa-cel was associated with veno-occlusive disease (ROR = 7.03) and hypoperfusion (ROR = 21.54). Axi-cel was associated with hypertensive emergency (ROR = 7.85) and deep vein thrombosis (ROR = 1.85). Ide-cel was found to be associated with hypertension (ROR = 2.40) (Table 3).

### 2.3. Clinical Characteristics of the Adverse Events

Among the AEs of the target, most AEs overlapped with CRS (overlap rate 49.6%-84.8%). The majority of AEs occurred within 7 days after infusion, with some cases of hypogammaglobulinemia and infections occurring later, more than 60 days after infusion, accounting for 11.2% and 18.1%, respectively, as shown in Figure 3 and Figure 4.

## 3. Discussion

The emergence of CAR T-cell therapy has expanded the therapeutic options available for treating hematological tumors. However, safety remains a paramount concern for both medical practitioners and patients when making treatment decisions involving these drugs. This study conducted a comparative analysis of the AE signals of Tisa-cel, Axi-cel, Brexu-cel, Liso-cel, Ide-cel, and Cilta-cel within the FAERS database. By examining both common and rare serious AEs, we obtained comprehensive insights into the safety profiles of these six drugs. These findings indicate that the toxicity profile of CAR T-cell therapy is relatively distinctive and potentially affects nearly every major organ system. Patients may experience toxicities in the respiratory, cardiovascular, hematological, renal, neurological, and gastrointestinal systems, ranging from mild to life-threatening. Furthermore, there are variations in AEs among different CAR T-cell products. While the detected AEs generally align with those reported in the drug labels, this study also identified some novel AE signals, warranting clinical attention.

Neurotoxicity is one of the most prevalent AEs in CAR T-cell therapy, with an incidence ranging from 21% to 66% [33,34,35], and strong signal intensity was detected in this study. In CD19 CAR T-cell therapy, ICANS ranges from altered consciousness to severe brain edema, while in BCMA CAR T-cell therapy, notable toxicity, such as parkinsonism, has occurred in pivotal clinical trials for multiple myeloma [24,25], consistent with the findings of this study. The on-target/off-tumor effect on BCMA+ cells in the basal ganglia is a potential mechanism. However, the reversibility and decline in frontal lobe metabolism suggest additional pathological mechanisms that necessitate further elucidation [36,37]. This parkinsonism, characterized by neurocognitive and dyskinetic features, is sparsely documented in published studies, contrasting with the neurotoxicity observed during CAR T-cell proliferation and cytokine release. Notably, this Parkinson-like neurotoxicity progresses and manifests nearly 3 months after the initiation of CAR T-cell infusion, warranting heightened clinical vigilance. Apart from monitoring CAR T-cell levels, clinicians can employ simple assessments, such as handwriting evaluations, during post-CAR T-cell monitoring. Any observed changes should prompt the consideration of proactive interventions, including steroid administration or alternative therapies, to mitigate effects prior to inducing brain injury. Signals of brain edema were also detected by Tisa-cel, Axi-cel, and Brexu-cel. Patients may rapidly transition from mild lethargy to unconsciousness within hours, culminating in fatality [35,38,39], underscoring the importance of vigilance.

CRS is the most common AE after CAR T-cell infusion, showing strong signal intensity across all six drugs evaluated in this study. The risk, severity, and duration of CRS subsequent to CAR T-cell therapy may be modulated by host-, tumor-, and/or treatment-related factors. Patients with a heightened tumor burden, intensified pretreatment, and elevated baseline levels of C-reactive protein, ferritin, D-dimer, and proinflammatory cytokines may face increased susceptibility to CRS and severe manifestations thereof. Moreover, variations exist in the incidence and severity of CRS across different CAR T-cell products. The incidence of moderate to severe CRS in Tisa-cel, Axi-cel, and Brexu-cel ranges from 8% to 48% [40,41,42], whereas the incidence of CRS in Liso-cel is relatively lower, at 1.1% to 9% [43]. With respect to BCMA, Ide-cel and Cilta-cel exhibit moderate to severe CRS incidence rates ranging from 3% to 9% [44,45]. Characterized by hypotension and hyperthermia, moderate to severe CRS may progress to shock, vascular leak syndrome, disseminated intravascular coagulation (DIC), and multiple organ dysfunction syndrome (MODS), posing life-threatening risks [46]. Continuous electrocardiogram (ECG) monitoring from CAR T-cell reinfusion onset is recommended, enabling early intervention to reverse CRS, including severe cases. Importantly, preventive measures against CRS do not compromise CAR T-cell functionality [47].

Patients receiving CAR T-cell therapy are often in advanced stages of disease, have received multiple lines of treatment, and require pretreatment chemotherapy before CAR T-cell infusion, which often results in severe bone marrow suppression, increasing the risk of infection. There are notable disparities in the pathogens responsible for infections within 30 days post-CAR T-cell infusion compared to those occurring after 30 days. The majority of infections occur within 30 days post-infusion, predominantly bacterial, followed by viral and fungal infections [48]. At 30 days post-infusion, the infection rate declines, but due to decreased immunoglobulin and lymphocytes, viral infections, including respiratory viruses, cytomegalovirus, herpes viruses, and Epstein–Barr virus, become predominant. In this study, a greater incidence of bacterial and viral infections was observed, consistent with previous research findings. The incidence of infection related to CAR T-cell therapy varies significantly across different primary diseases, CAR T-cell therapy products, and medical centers [33]. Signal comparison revealed that compared to other drugs, Cilta-cel poses a relatively greater risk of bacterial, viral, and fungal infections, suggesting that patients receiving Cilta-cel treatment are more susceptible to infections. William et al. reported that infection is a leading cause of non-relapse mortality (NRM) among multiple myeloma patients receiving CAR T-cell therapy [49]. Hence, infection prevention and control are paramount in CAR T-cell therapy and warrant anti-infective treatment according to relevant guidelines. Additionally, attention should be given to patients’ immunoglobulin levels, with intravenous immunoglobulin supplementation administered when necessary [50,51].

Bone marrow suppression associated with CAR T-cell therapy ranks among the common complications in patients with hematological malignancies, characterized by leukopenia, anemia, or thrombocytopenia, often with multiline involvement and a higher incidence of AEs above grade 3. Approximately 30% of patients experience a recovery period exceeding one month [52]. Signal comparison indicated that compared with other drugs, Tisa-cel has the highest risk of hematological toxicity [40,41,42,43,44,45]. High-dose pretreatment can lead to decreased blood cell counts post-CAR T-cell therapy, with severe CRS, infections, high tumor burden, and low baseline blood cell levels contributing to delayed hematopoietic recovery. Although some patients experience prolonged recovery times for blood cells, they do not necessarily develop serious delayed complications and thus do not require specific treatment [17]. However, increased attention should be given to the heightened infection risk during periods of blood cell decline, with hematopoietic growth factors and blood product transfusions administered as necessary [53,54]. Approximately 90% of patients receiving CAR T-cell therapy exhibit at least one abnormal coagulation index, manifested by increased D-dimer levels, decreased blood fibrinogen levels, prolonged activated partial thromboplastin time, and prolonged prothrombin time [55]. In this study, Tisa-cel and Axi-cel exhibited strong signal intensities, while other drugs did not, likely due to their later market introduction and limited data availability. Current understanding suggests that abnormal coagulation function is primarily related to severe CRS and CAR-HLH. It is crucial to note that thrombocytopenia combined with abnormal coagulation can exacerbate bleeding risk, potentially resulting in life-threatening organ bleeding (e.g., cerebral hemorrhage, alveolar hemorrhage, and gastrointestinal bleeding), necessitating close clinical monitoring.

In addition to common AEs, this study analyzed rare but life-threatening AEs. Previous research has demonstrated that CAR T-cell therapy may induce various cardiorespiratory toxicities with high mortality rates [56,57,58,59,60]. Our study revealed a broad spectrum of cardiorespiratory toxicity, with 39 signals disproportionately associated with CAR T-cell therapy, indicating multifaceted cardiorespiratory toxicity related to CAR T-cell therapy. Understanding the diversity in toxicities and the potential risks associated with these drugs is crucial for enhancing toxicity management, representing one of the most important avenues for overall improvement in CAR T-cell therapy. Notably, signals such as hypoxia, tachypnoea, cardiorenal syndrome, and hypotension stand out due to their strong disproportionality. Moreover, some toxicities, such as atelectasis and shock, are underrepresented in published data or remain inadequately recognized. By presenting all potential signals and quantifying their risks, we offer a comprehensive and systematic insight into these rare toxicities, which is a crucial complement to current pharmacovigilance research. Notably, there was a significant overlap between CAR T-cell-related CRS and CAR T-cell-related cardiorespiratory toxicity, consistent with previous findings [56,58,59,61]. However, the underlying pathological mechanisms of heart injury associated with this novel therapy remain unclear. Patients with evident or underlying cardiovascular diseases may be more susceptible to CAR T-cell-related cardiotoxicity [62]. Therefore, a detailed cardiovascular history and physical examination should be conducted before treatment initiation. A retrospective study at Massachusetts General Hospital revealed that cardiovascular events occurred only in patients with grade ≥ 2 CRS, with longer durations between CRS onset and tocilizumab administration associated with higher cardiovascular event risks, increasing by 1.7 times every 12 h. Early tocilizumab use for CRS treatment can mitigate the risk of cardiovascular AEs [56]. Mahmood et al. also reported that combined severe cardiovascular events, such as heart failure, cardiogenic shock, or myocardial infarction, increase the risk of death and correlate with increased peak levels of IL-6, CRP, ferritin, and troponin. Clinicians should remain vigilant regarding these rare but potentially fatal cardiovascular toxicities following CAR T-cell therapy. Monitoring, including continuous echocardiography, baseline biomarkers, and electrocardiograms, should be intensified during treatment.

This study has certain limitations. First, the research data were sourced from a spontaneous reporting system (SRS); underreporting is one of the most important limitations on SRS, as it prevents the precise calculation of the real incidence of the event in the population. In fact, only a small proportion of the AEs that occur in daily practice are reported [63]. However, large-scale data mining still contributes to warning about the safe and rational use of drugs. Second, the reports in the FAERS database predominantly originated from European and American countries, with a relatively low proportion from Asian populations, potentially resulting in geographical bias in the results. Locke et al.’s study [64] indicated that ICANS might vary by race, and we plan to conduct retrospective or prospective studies incorporating more Asian populations to achieve a more comprehensive analysis of AEs. Third, all signal detection results can only indicate potential associations between drugs and adverse events; they cannot establish a definitive causal relationship between drugs and adverse events. In the future, we will conduct prospective observational studies or clinical trials to determine the causal relationship between CAR T-cell therapy and AEs. Future research combining the FAERS database with other data sources is crucial for continuous monitoring of CAR T-cell-related toxicity.

## 4. Materials and Methods

### 4.1. Study Design and Data Source

This was a real-world, observational, retrospective pharmacovigilance study, and the data were obtained from the FAERS database. Six CAR T-cell products approved by the FDA (tisagenlecleucel, axicabtagene ciloleucel, brexucabtagene autoleucel, lisocabtagene maraleucel, idecabtagene vicleucel, and ciltacabtagene autoleucel) were selected as the study drugs. All files are available on the FDA website “https://fis.fda.gov/extensions/fpd-qde-faers/fpd-qde-faers.html (accessed on 25 April 2024)”. In this study, all data from 26 quarters of the ASCII packets, spanning from the fourth quarter of 2017 to the first quarter of 2024, were extracted and imported into SAS 9.4 software for data cleaning and analysis. Since the FAERS is a publicly available and anonymized database, neither informed consent nor ethical approval was involved.

### 4.2. Data Extraction and Mining

The data extraction and mining processes used in this study are shown in Figure 1. The target population was screened based on DRUGNAME and PROD_AI, with the degree of reporting suspicion limited to “primary suspected drug (PS)”. According to the FDA’s recommended method for eliminating duplicate reports, the fields PRIMARYID, CASEID, and FDA_DT from the DEMO table were selected and sorted by CASEID, FDA_DT, and PRIMARYID. For reports with the same CASE number, the latest (most recent) FDA_DT was selected. For those occasions when both the CASE and FDA_DT fields were the same, the report with the higher ISR was selected.

Since the FAERS database records AE names using the preferred terms (PT) terminology of the *Medical Dictionary for Regulatory Activities* (MedDRA), and the MedDRA is updated every March and September, involving adjustments to PT levels and changes to system organ classes (SOCs), the latest version of the MedDRA was used to correct the PT names in the FAERS database. The SOC and PT from the latest version of the MedDRA were obtained. Clinical characteristics, including demographic features (age and sex), reporting characteristics (reporting year, region, and reporter’s occupation), and reporting indications, were also collected. Additionally, the time to onset of specific AEs was also assessed, calculated as the interval between the start date of medication (START_DT) and the time of occurrence of AEs (EVENT_DT). Reports with missing or incorrect data (drug usage time later than the time of event occurrence) were excluded.

### 4.3. Statistical Analysis

The measure of disproportionality is the main method used for research on AE signals and is based on the two-by-two contingency table (Appendix A). Its principle is to compare the difference between the frequency of the target drug event and the background frequency. When the frequency of the target drug event is significantly higher than the background frequency and exceeds the set threshold, it is called disproportionality, indicating that some connection may exist between the suspicious drugs and suspected AEs, not due to chance factors or the “noisy background” of the database [65,66,67].

The disproportionality analysis was conducted using the reporting odds ratio (ROR) method and the Medicines and Healthcare Products Regulatory Agency (MHRA) method, which have high reliability and sensitivity in the proportional imbalance method. The study used the ROR and its confidence interval and the proportional reporting ratio (PRR) and its χ^2^ value to detect the AE signal. To avoid the appearance of false positive signals, the data should meet the number of AE reports ≥ 3, the 95% CI lower limit of ROR > 1, PRR > 2, and χ^2^ > 4, further excluding reports that did not meet the requirements [68]. The signal intensities were arranged in descending order of the ROR value. The larger the ROR is, the stronger the signal, indicating a stronger correlation between the drug and the AE. All data processing and statistical analysis were performed using SPSS version 26.0 and Prism version 10.1.

## 5. Conclusions

CRS and ICANS are the most common AEs associated with CAR T-cell therapy. Apart from these, additional AEs may include infection, cytopenia, hypogammaglobulinemia, HLH, coagulation dysfunction, and hypotension. Furthermore, there were variations in the AE spectra among the different CAR T-cell products. Clinicians should assess patients prior to medication and closely monitor their vital signs, mental status, and laboratory parameters during treatment. Prompt intervention should be implemented if AEs occur, thereby effectively reducing the clinical risks associated with medication.

## Figures and Tables

**Figure 1 pharmaceuticals-17-01025-f001:**
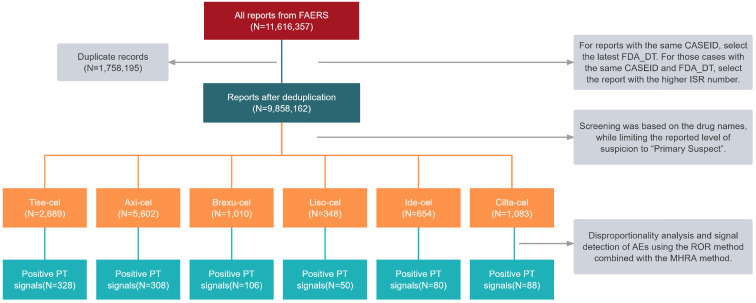
Flow diagram for data mining from the FAERS database. Positive PT signals: positive signals at the preferred term (PT) level.

**Figure 2 pharmaceuticals-17-01025-f002:**
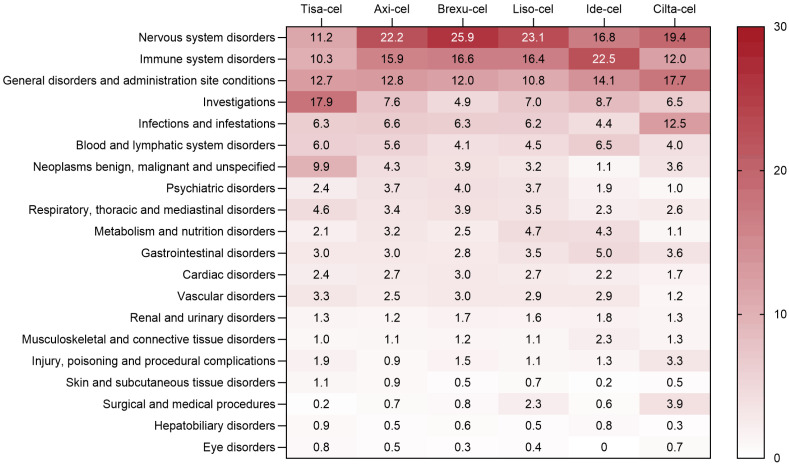
Distribution of adverse event reporting rates.

**Figure 3 pharmaceuticals-17-01025-f003:**
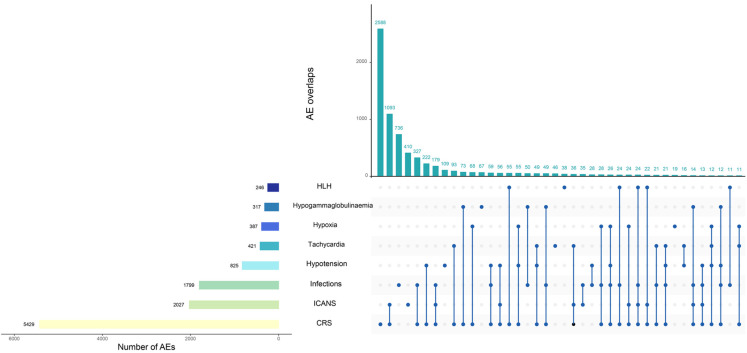
Overlap within major adverse events.

**Figure 4 pharmaceuticals-17-01025-f004:**
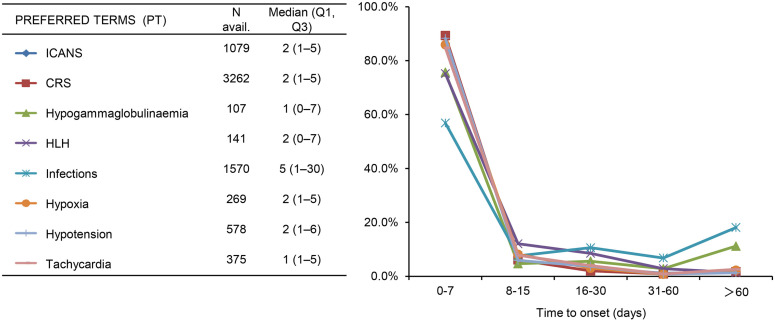
Days from CAR T-cell infusion to adverse event onset.

**Table 1 pharmaceuticals-17-01025-t001:** Characteristics of reports associated with CAR T-cell therapies.

Variable (n, %)	Tisa-cel	Axi-cel	Brexu-cel	Liso-cel	Ide-cel	Cilta-cel
No. of patients	2689	5602	1010	348	654	1083
Sex						
Female	835 (31.1)	1734 (31.0)	206 (20.4)	127 (36.5)	231 (35.3)	273 (25.2)
Male	1297 (48.2)	2716 (48.5)	630 (62.4)	208 (59.8)	326 (49.8)	364 (33.6)
Not specified	557 (20.7)	1152 (20.6)	174 (17.2)	13 (3.7)	97 (14.8)	446 (41.2)
Age, years						
<18	578 (21.5)	16 (0.3)	8 (0.8)	1 (0.3)	0 (0.0)	0 (0.0)
18–64	701 (26.1)	2323 (41.5)	341 (33.8)	99 (28.4)	198 (30.3)	144 (13.3)
≥65	458 (17.0)	1522 (27.2)	344 (34.1)	218 (62.6)	307 (46.9)	205 (18.9)
Not specified	952 (35.4)	1741 (31.1)	317 (31.4)	30 (8.6)	149 (22.8)	734 (67.8)
Reporting year						
2017	9 (0.3)	1 (0.0)	-	-	-	-
2018	135 (5.0)	479 (8.6)	-	-	-	-
2019	302 (11.2)	785 (14.0)	-	-	-	-
2020	601 (22.4)	861 (15.4)	25 (2.5)	-	-	-
2021	535 (19.9)	678 (12.1)	206 (20.4)	86 (24.7)	94 (14.4)	-
2022	556 (20.7)	898 (16.0)	283 (28.0)	124 (35.6)	264 (40.4)	161 (14.9)
2023	424 (15.8)	1313 (23.4)	362 (35.8)	104 (29.9)	218 (33.3)	607 (56.0)
2024	127 (4.7)	587 (10.5)	134 (13.3)	34 (9.8)	78 (11.9)	315 (29.1)
Reporter						
Health professional	2284 (84.9)	4820 (86.0)	831 (82.3)	240 (69.0)	459 (70.2)	569 (52.5%)
Consumer	331 (12.3)	393 (7.0)	58 (5.7)	29 (8.3)	84 (12.8)	463 (42.8)
Not specified	74 (2.8)	389 (6.9)	121 (12.0)	79 (22.7)	111 (17.0)	51 (4.7)
Reporting region						
North America	1767 (65.7)	3421 (61.1)	574 (56.8)	177 (50.9)	405 (61.9)	878 (81.1)
Europe	476 (17.7)	1427 (25.5)	247 (24.5)	15 (4.3)	81 (12.4)	72 (6.6)
Asia	207 (7.7)	219 (3.9)	7 (0.7)	32 (9.2)	19 (2.9)	16 (1.5)
Not specified/other	239 (8.9)	535 (9.6)	182 (18.0)	124 (35.6)	149 (22.8)	117 (10.8)
Indication						
Acute lymphoblastic leukemia	1054 (39.2)	31 (0.6)	161 (15.9)	1 (0.3)	-	-
Large B-cell lymphoma	1004 (37.3)	3256 (58.1)	16 (1.6)	237 (68.1)	3 (0.5)	2 (0.2)
Follicular lymphoma	49 (1.8)	148 (2.6)	-	17 (4.9)	-	-
Mantle cell lymphoma	5 (0.2)	39 (0.7)	517 (51.2)	7 (2.0)	-	-
Multiple myeloma	-	-	-	-	578 (88.4)	417 (38.5)
Not specified/other	577 (21.5)	2128 (38.0)	316 (31.3)	86 (24.7)	73 (11.2)	664 (61.3)
AE Severity						
Serious	2499 (92.9)	5394 (96.3)	955 (94.6)	309 (88.8)	572 (87.5)	759 (70.1)
Non-serious	190 (7.1)	208 (3.7)	55 (5.4)	39 (11.2)	82 (12.5)	324 (29.9)
Outcome						
Life-threatening	273 (10.2)	361 (6.4)	102 (10.1)	31 (8.9)	36 (5.5)	46 (4.2)
Hospitalization	831 (30.9)	2011 (35.9)	405 (40.1)	205 (58.9)	241 (36.9)	402 (37.1)
Disability	42 (1.6)	82 (1.5)	18 (1.8)	2 (0.6)	7 (1.1)	4 (0.4)
Death	707 (26.3)	1322 (23.6)	247 (24.5)	61 (17.5)	60 (9.2)	127 (11.7)
Other	1821 (67.7)	4536 (81.0)	744 (73.7)	140 (40.2)	396 (60.6)	348 (32.1)

**Table 2 pharmaceuticals-17-01025-t002:** Signal detection of common adverse events in key organs associated with CAR T-cell therapies at the preferred terms (PT) level.

Preferred Terms (PT)	Tisa-cel		Axi-cel		Brexu-cel		Liso-cel		Ide-cel		Cilta-cel	
	N	ROR (95% CI)	N	ROR (95% CI)	N	ROR (95% CI)	N	ROR (95% CI)	N	ROR (95% CI)	N	ROR (95% CI)
Nervous system disorders												
ICANS	268	223.33 (196.65–253.63)	1202	1198.80 (1109.64–1295.11)	324	801.69 (709.79–905.49)	64	426.30 (329.95–550.79)	109	313.35 (257.36–381.52)	60	178.74 (137.74–231.96)
Neurotoxicity	288	63.36 (56.29–71.33)	1156	209.88 (197.02–223.57)	151	145.20 (123.07–171.29)	72	236.75 (186.02–301.32)	55	79.03 (60.38–103.44)	43	71.22 (52.55–96.52)
Encephalopathy	67	12.40 (9.75–15.77)	247	33.92 (29.88–38.51)	45	36.13 (26.90–48.54)	6	16.10 (7.21–35.94)	7	8.52 (4.06–17.90)	6	8.43 (3.78–18.79)
Aphasia	48	7.64 (5.75–10.14)	204	24.13 (21.00–27.73)	33	24.35 (17.27–34.33)	14	35.22 (20.77–59.72)	5	5.54 (2.30–13.33)	3	3.89 (1.25–12.07)
Tremor ^a^	-	-	198	4.28 (3.72–4.92)	45	6.26 (4.67–8.40)	11	5.16 (2.85–9.34)	27	5.74 (3.93–8.39)	-	-
Depressed level of consciousness	43	5.06 (3.75–6.82)	56	4.77 (3.67–6.20)	17	9.05 (5.61–14.57)	5	8.90 (3.74–21.67)	18	23.97 (15.06–38.16)	-	-
Dysgraphia	16	11.30 (6.91–18.48)	56	32.33 (24.80–42.14)	8	29.24 (14.58–58.64)	-	-	-	-	-	-
Apraxia	3	9.43 (3.03–29.32) *	3	6.80 (2.19–21.14)	-	-	-	-	-	-	-	-
Brain oedema	18	7.13 (4.49–11.33)	43	12.51 (9.26–16.90)	14	24.55 (14.50–41.54)	-	-	-	-	3	8.95 (2.88–27.80)
Cerebellar infarction	4	13.75 (5.14–36.76)	6	15.17 (6.79–33.92)	-	-	-	-	-	-	-	-
Brain stem infarction	-	-	3	14.69 (4.71–45.83)	-	-	-	-	-	-	-	-
Embolic stroke	-	-	5	5.05 (2.10–12.16)	-	-	-	-	-	-	-	-
Intracranial hemorrhage	-	-	-	-	-	-	-	-	-	-	5	18.47 (7.67–44.48) *
Cerebral hemorrhage	-	-	-	-	-	-	-	-	-	-	3	3.81 (1.23–11.82) *
Cranial nerve disorder	3	19.93 (6.39–62.16) *	-	-	-	-	-	-	-	-	8	991.24 (467.81–2100.39)
Facial nerve disorder	-	-	-	-	-	-	-	-	-	-	6	727.22 (310.87–1701.16)
Bell’s palsy	-	-	-	-	-	-	-	-	-	-	33	253.07 (178.12–359.57)
Facial paralysis ^b^	22	6.24 (4.11–9.49) *	-	-	4	5.56 (2.08–14.83) *	-	-	-	-	35	95.69 (68.32–134.02)
Parkinsonism	-	-	-	-	-	-	-	-	4	13.00 (4.87–34.72)	43	168.90 (124.33–229.44)
Motor dysfunction	12	5.40 (3.06–9.52)					-	-	-	-	-	-
Optic neuritis	6	3.57 (1.60–7.94)	-	-	-	-	-	-	-	-	-	-
Guillain–Barre syndrome	4	3.81 (1.43–10.15) *	-	-	-	-	-	-	-	-	3	19.92 (6.41–61.92)
Cerebellar syndrome	3	7.82 (2.52–24.31) *	3	5.92 (1.90–18.39) *	-	-	-	-	-	-	-	-
Dysmetria	-	-	3	10.30 (3.31–32.07) *	-	-	-	-	-	-	-	-
Balance disorder	-	-	-	-	-	-	-	-	18	6.62 (4.16–10.53)	-	-
Cerebral mass effect	-	-	3	10.58 (3.40–32.96) *	-	-	-	-	-	-	-	-
Cerebral venous sinus thrombosis	-	-	4	8.70 (3.25–23.25)	-	-	-	-	-	-	-	-
Paraplegia	-	-	6	7.17 (3.21–15.98) *	-	-	-	-	-	-	-	-
Myelopathy	-	-	4	6.00 (2.25–16.03) *	-	-	-	-	-	-	-	-
Ageusia	-	-	-	-	-	-	-	-	-	-	4	5.22 (1.96–13.94) *
Immune system disorders												
CRS ^c^	1139	269.77 (253.08–287.57)	2965	676.42 (646.44–707.79)	539	421.97 (383.85–463.88)	153	369.83 (311.01–439.77)	448	515.97 (464.10–573.64)	198	213.05 (183.57–247.27)
Hypogammaglobulinemia	213	151.64 (131.80–174.47)	65	31.74 (24.81–40.60)	-	-	4	33.71 (12.61–90.10) *	30	113.34 (78.80–163.02)	4	16.28 (6.09–43.47)
HLH ^d^	78	30.82 (24.63–38.57)	87	24.99 (20.21–30.91)	23	34.84 (23.08–52.57)	11	54.44 (30.02–98.72)	17	37.52 (23.25–60.54)	30	75.22 (52.33–108.13)
Graft versus host disease ^e^	21	6.51 (4.24–10.00)	7	5.67 (2.70–11.91) *	-	-	-	-	-	-	-	-
Immunodeficiency	-	-	6	133.41 (57.74–308.26) *	-	-	-	-	22	32.43 (21.29–49.41)	3	4.96 (1.60–15.41)
Infections and infestations												
Bacterial infections ^f^	244	4.59 (4.04–5.20)	273	3.47 (3.08–3.91)	36	3.69 (2.66–5.12)	9	8.30 (4.30–16.00)	32	7.82 (5.51–11.08)	40	7.16 (5.23–9.79)
Fungal infections ^f^	85	3.14 (2.53–3.88)	133	3.61 (3.04–4.28)	26	4.94 (3.36–7.26)	4	5.52 (2.07–14.74)	10	4.92 (2.64–9.15)	17	8.17 (5.07–13.18)
Viral infections ^f^	-	-	-	-	-	-	-	-	-	-	79	2.63 (2.10–3.30)
Investigations												
White blood cell count decreased	229	8.63 (7.57–9.84)	118	3.19 (2.66–3.82)	-	-	6	2.96 (1.33–6.60)	21	4.61 (3.00–7.09)	8	2.02 (1.01–4.05)
Platelet count decreased	232	9.33 (8.19–10.62)	111	3.21 (2.66–3.86)	-	-	10	5.26 (2.82–9.80)	-	-	13	3.49 (2.02–6.03)
Hemoglobin decreased	222	10.44 (9.14–11.92)	-	-	-	-	-	-	-	-	-	-
Neutrophil count decreased	198	20.43 (17.75–23.52)	124	9.15 (7.66–10.92)	-	-	-	-	3	26.21 (8.43–81.53)	5	3.09 (1.28–7.43)
Aspartate aminotransferase increased	30	3.30 (2.30–4.72)	95	7.63 (6.24–9.34) *	22	10.31 (6.78–15.69)	3	4.91 (1.58–15.26) *	47	34.99 (26.19–46.75)	9	7.50 (3.90–14.45)
Alanine aminotransferase increased	29	2.49 (1.73–3.58)	78	4.86 (3.89–6.07) *	23	4.71 (3.13–7.10)	-	-	42	24.46 (18.01–33.20)	8	5.31 (2.65–10.63)
Blood fibrinogen decreased	24	112.61 (74.61–169.94)	21	70.43 (45.42–109.19)	3	55.78 (17.87–174.15)	-	-	-	-		
Serum ferritin increased	84	79.22 (63.66–98.59)	71	48.25 (38.07–61.15) *	13	56.30 (32.56–97.36) *	6	90.51 (40.46–202.43) *	-	-	3	21.70 (6.98–67.49)
Activated partial thromboplastin time prolonged	19	30.32 (19.27–47.72) *	-	-	-	-	-	-	-	-	-	-
Prothrombin time prolonged	16	24.03 (14.67–39.35)	-	-	-	-	-	-	-	-	-	-
Fibrin D dimer increased	15	18.49 (11.12–30.75)	-	-	-	-	-	-	-	-	-	-

^a^ Tremor includes tremor, intention tremor, dystonic tremor, resting tremor, essential tremor, action tremor, and postural tremor. ^b^ Facial paralysis includes facial paralysis and facial paresis. ^c^ Cytokine release syndrome includes cytokine release syndrome and cytokine storm. ^d^ HLH includes HLH and immune effector cell-associated HLH-like syndrome. ^e^ Graft versus host disease includes graft versus host disease, acute graft versus host disease, chronic graft versus host disease, graft versus host disease in liver, and graft versus host disease in skin. ^f^ Infections and infestations: system organ class adverse events are grouped by high-level group term (HLGT) pathogen type. * Adverse events not mentioned on FDA drug labels.

**Table 3 pharmaceuticals-17-01025-t003:** Signal detection of adverse events of special interest associated with CAR T-cell therapies at the preferred terms (PT) level.

Preferred Terms (PT)	Tisa-cel		Axi-cel		Brexu-cel		Liso-cel		Ide-cel		Cilta-cel	
	N	ROR (95% CI)	N	ROR (95% CI)	N	ROR (95% CI)	N	ROR (95% CI)	N	ROR (95% CI)	N	ROR (95% CI)
Respiratory, thoracic, and mediastinal disorders												
Hypoxia	167	21.53 (18.47–25.09)	171	15.95 (13.71–18.55)	32	17.24 (12.16–24.43)	4	7.32 (2.74–19.55)	11	9.22 (5.10–16.68)	-	-
Respiratory failure ^a^	72	3.68 (2.92–4.64)	74	2.76 (2.19–3.47)	20	4.71 (3.03–7.31)	4	12.27 (4.59–32.76)	-	-	15	6.65 (4.00–11.06)
Tachypnoea	38	12.19 (8.86–16.77)	37	8.59 (6.21–11.87)	12	15.80 (8.96–27.87)	-	-	5	9.88 (4.10–23.77)	3	6.87 (2.21–21.33)
Pleural effusion ^b^	36	2.79 (2.01–3.87)	70	3.97 (3.14–5.02)	15	5.40 (3.25–8.98)	7	8.81 (4.19–18.52)	6	3.36 (1.51–7.48) *	-	-
Atelectasis	12	7.87 (4.46–13.88) *	6	2.85 (1.28–6.35) *	4	11.27 (4.22–30.08) *	-	-	-	-	-	-
Respiratory distress	38	4.08 (2.96–5.61)	17	3.29 (2.05–5.30) *	-	-	-	-	-	-	3	6.14 (1.98–19.06) *
Pulmonary oedema	23	2.46 (1.63–3.70)	-	-	-	-	-	-	-	-	-	-
Pulmonary hemorrhage	8	5.29 (2.64–10.59)	8	3.87 (1.93–7.75)	-	-	-	-	-	-		
Pulmonary mass	8	2.05 (1.02–4.10) *	-	-	-	-	-	-	-	-	-	-
Lung consolidation	8	5.99 (2.99–11.98) *	3	4.07 (1.31–12.65) *	-	-	-	-	-	-	-	-
Organizing pneumonia	6	5.00 (2.24–11.14) *	-	-	-	-	-	-	-	-	-	-
Apnea	4	2.84 (1.07–7.58) *	-	-	-	-	-	-	-	-	-	-
Laryngeal oedema	4	3.40 (1.28–9.08) *	-	-	-	-	-	-	5	30.63 (12.71–73.82)	-	-
Pharyngeal hemorrhage	3	10.64 (3.42–33.10)	-	-	-	-	-	-	-	-	-	-
Aspiration	-	-	12	4.19 (2.38–7.39) *	-	-	-	-	-	-	-	-
Lung infiltration	-	-	6	3.56 (1.60–7.94) *	-	-	-	-	-	-	-	-
Pulmonary alveolar hemorrhage	-	-	5	2.98 (1.24–7.16)	-	-	-	-	-	-	-	-
Cardiac disorders												
Tachycardia ^c^	143	5.30 (4.50–6.25)	201	5.44 (4.74–6.26)	47	7.84 (5.88–10.46)	5	3.34 (1.39–8.04)	29	7.82 (5.42–11.28)	8	2.99 (1.49–5.99)
Arrhythmia	94	1.23 (1.01–1.51)	200	1.856 (1.61–2.126)	37	2.31 (1.67–3.20)	17	4.45 (2.75–7.18)	18	1.75 (1.10–2.79)	-	-
Cardiorenal syndrome	8	44.41 (22.03–89.50) *	35	150.61 (106.25–213.50) *	-	-	-	-	-	-	-	-
Pericardial effusion	12	2.41 (1.37–4.25) *	-	-	-	-	-	-	-	-	-	-
Left ventricular dysfunction	8	5.27 (2.63–10.54)	-	-	-	-	-	-	-	-	-	-
Mitral valve disease	8	3.78 (1.89–7.56) *	-	-	-	-	-	-	-	-	-	-
Aortic valve incompetence	3	4.71 (1.52–14.64) *	-	-	-	-	-	-	-	-	-	-
Cardiomyopathy	-	-	25	4.43 (3.00–6.57) *	-	-	-	-	-	-	-	-
Cardiopulmonary failure	-	-	4	4.41 (1.65–11.77) *	-	-	-	-	-	-	-	-
Vascular disorders												
Hypotension ^d^	335	6.98 (6.27–7.78)	346	5.22 (4.69–5.81)	76	7.12 (5.67–8.94)	26	8.15 (5.52–12.03)	41	6.21 (4.56–8.46)	16	2.76 (1.69–4.51)
Shock ^e^	22	3.20 (2.11–4.87) *	32	2.87 (2.03–4.06) *	7	4.31 (2.05–9.06) *	-	-	3	4.41 (1.42–13.70) *	-	-
Hemodynamic instability	9	5.23 (2.72–10.06) *	9	3.75 (1.95–7.22) *	-	-	-	-	-	-	-	-
Circulatory collapse	7	2.21 (1.05–4.64) *	-	-	-	-	-	-	-	-	-	-
Veno-occlusive disease	5	7.03 (2.92–16.91) *	-	-	-	-	-	-	-	-	-	-
Hypoperfusion	4	21.54 (8.04–57.69) *	-	-	-	-	-	-	-	-	-	-
Ischemia	3	3.80 (1.22–11.79 *	-	-	-	-	-	-	-	-	-	-
Hypertensive emergency	-	-	3	7.85 (2.52–24.43) *	-	-	-	-	-	-	-	-
Deep vein thrombosis	-	-	25	1.85 (1.25–2.74)	-	-	-	-	-	-	-	-
Hypertension	-	-	-	-	-	-	-	-	18	2.40 (1.51–3.82)	-	-

^a^ Respiratory failure includes respiratory failure and acute respiratory failure. ^b^ Pleural effusion includes pleural effusion and malignant pleural effusion. ^c^ Tachycardia includes tachycardia, sinus tachycardia, supraventricular tachycardia, and ventricular tachycardia. ^d^ Hypotension includes hypotension, capillary leak syndrome, and orthostatic hypotension. ^e^ Shock includes shock, hemorrhagic shock, distributive shock, hypovolemic shock, and neurogenic shock. * Adverse events not mentioned on FDA drug labels.

## Data Availability

The FDA Adverse Event Reporting System data are available at https://fis.fda.gov/extensions/fpd-qde-faers/fpd-qde-faers.html (accessed on 26 April 2024).

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
