# Peer review of "Comparing the Differences in Adverse Events among Chimeric Antigen Receptor T-Cell Therapies: A Real-World Pharmacovigilance Study"

_pharmaceuticals, 2024, doi:10.3390/ph17081025_

Round 1

Reviewer 1 Report

Comments and Suggestions for Authors

The research manuscript compares adverse events (AEs) among FDA-approved CAR T-cell products by analyzing data submitted to the FDA Adverse Event Reporting System (FAERS). The article discusses the most common AEs, as well as AEs that occur with lower frequency and ones which are not indicated in the description of medications but warrant clinical attention. The authors emphasize that the study has certain limitations and that further research combining the FAERS database with other data sources is required.

The manuscript is well-written, but may benefit from these revisions:

1. Describe what is meant by "Positive PT signals" in Figure 1.

2. Please remove the abbreviation MM and use "Multiple myeloma" instead in Table 1.

3. Format Table 2 so that the number of cases (N) is written on one line.

4. In Table 2, you label adverse events with "*", such as Guillain-Barré syndrome, increased serum ferritin, and increased alanine aminotransferase, indicating that they are not mentioned in the FDA drug labeling of Cilta-cel (Carvykti). However, these side effects are indicated in the prescription of Carvykti. Also, increased serum ferritin (hyperferritinemia) is mentioned in the drug label of Tisa-cel (Kymriah). Please review and correct the information.

5. On page 13, the authors write: "Signal comparison revealed that compared to 238 other drugs, Cilta-cel poses a relatively greater risk of bacterial, viral, and fungal infections, suggesting that patients receiving Cilta-cel treatment are more susceptible to infections." The authors refer to an article in which Los-Arcos et al. reported that infection is a leading cause of nonrelapse mortality among DLBCL patients receiving CAR T-cell therapy in standard nursing environments. However, Cilta-cel is used for MM treatment. Please revise your citation and choose the correct article.

6. Figure 2: I would suggest placing drug labels at the top of the figure.

7. Format Figure 3, it is hard to see anything.

8. Please correct the typing errors in the manuscript:

   - Line 92: Include a hyphen in the name of the drug Ide-cel.

   - Line 98: Write "Table 2" in the singular.

   - Lines 95, 172, 174: Place a space after "Figure".

Reviewer 2 Report

Comments and Suggestions for Authors

This manuscript titled as “Manuscript Review: Comparing the differences in adverse events among CAR T-cell therapies: a real-world, pharmacovigilance study” aims to compare adverse events (AEs) among different CAR T-cell therapies using data from the FDA Adverse Event Reporting System (FAERS), providing cetain insights for clinical safety. Authors utilized FAERS data from Q4 2017 to Q1 2024, which includes a large number of AE reports, enhancing the robustness of the findings. This manuscript provides a comprehensive analysis of various AEs, categorizing them by system organ class (SOC) and specific adverse events, which helps in understanding the safety profiles of different CAR T-cell therapies. This manuscript contains certain clinical significance and provide a new approach for database-based research. With proper revision, acceptance could be considered. Main issues have been listed as following.

1.The majority of reports in the FAERS database originate from North America and Europe, with relatively few from Asia, which could introduce geographical bias. In addition, the demographic data, such as age and sex distribution, are not fully explored, which could impact the understanding of AE variations among different patient groups. Authors should point out and discuss such limitation in the manuscript. In other words, the reliance on a spontaneous reporting database like FAERS may lead to underreporting and potential bias in the quality and completeness of the data. Can authors add more data from other regions, especially Asia, to reduce geographical bias and enhance the generalizability of the findings?

2.The study design does not allow for the calculation of AE incidence rates due to the nature of the FAERS data, limiting the ability to determine the actual risk associated with each CAR T-cell product. Such issue should also be comprehensively discussed in the manuscript.

3.Signal detection methods used (ROR and MHRA) only suggest statistical correlations and do not establish causality. How to necessitate further research to confirm the findings? It will be better if authors conduct further studies to establish causality between CAR T-cell therapies and identified AEs, potentially through prospective observational studies or clinical trials.

4.This study does not thoroughly address potential confounding factors such as prior treatments, disease severity, and concomitant medications, which could influence the AE profiles.

Round 2

Reviewer 2 Report

Comments and Suggestions for Authors

This revised manuscript titled as "Comparing the differences in adverse events among CAR T-cell therapies: a real-world, pharmacovigilance study" uses a large dataset from the FAERS database, spanning multiple years, to analyze adverse events (AEs) related to various CAR T-cell products. 

Issues have been observed in this manuscript and some of them can’t be fixed. For instance, the manuscript relies on the FAERS database, which is subject to underreporting and potential biases in data quality and completeness, limiting the ability to calculate the true incidence of AEs. Additionally, the majority of reports in the FAERS database come from North America and Europe, with relatively few from Asia, leading to geographical bias that might affect the generalizability of the findings to other populations. The study includes data up to the first quarter of 2024, but the continuously evolving nature of CAR T-cell therapies and AE reporting might mean that newer AEs or changes in incidence rates are not captured. While the study provides a detailed statistical analysis of AEs, it may benefit from a deeper discussion on the clinical significance and impact of these findings on patient care. The manuscript highlights the need for monitoring but could offer more detailed recommendations on specific management strategies for the identified AEs. Lastly, the manuscript contains several typographical errors and formatting issues, such as inconsistent use of capitalization (e.g., "Immune system disorders" vs. "immune system disorders"), which detract from the overall readability.

However, this manuscript compares AEs across six different CAR T-cell products, providing certain insights into the differences and similarities in their safety profiles. The findings do have direct implications for clinicians, highlighting the need for careful monitoring and management of patients undergoing CAR T-cell therapy. Acceptance can be considered.